# Dementia in Media Coverage: A Comparative Analysis of Two Online Newspapers across Time

**DOI:** 10.3390/ijerph181910539

**Published:** 2021-10-08

**Authors:** Atiqur sm-Rahman, Chih Hung Lo, Yasmin Jahan

**Affiliations:** 1Division Ageing and Social Change, Department of Culture and Society, Linköping University, S-60174 Norrköping, Sweden; 2Department of Neurology, Brigham and Women’s Hospital, Harvard Medical School, Boston, MA 02115, USA; chihhunglo88@gmail.com; 3Graduate School of Biomedical and Health Sciences, Hiroshima University, Hiroshima 739-8527, Japan; dr.yasminjahan@gmail.com

**Keywords:** dementia, media portrayal, discourse, online newspapers, culture and society

## Abstract

The news media, specifically online newspapers, is one of the powerful transmitters of discourse due to its rapid accessibility that contributes to social beliefs and attitudes that often shape our perceptions on dementia and Alzheimer’s disease. The media portrayal of dementia is largely heterogeneous, but there is certainly an association between the influence of online news coverage and the social perceptions of dementia that need to be understood more broadly. In this study, we examined the portrayal of dementia in two online newspapers (*The New York Times* and *The Guardian*) that might have an influence on dementia discourse by comparing the content and form of the news coverage on dementia across time. This study was guided by three interconnected theoretical understandings: *cultivation theory, agenda-setting theory,* and *spiral of silence theory*. A total of 291 published articles featuring dementia from 2014 to 2019 were included in this study and a content analysis of the articles provided insight into the dementia-related news coverage. Our results showed that both newspapers have a decreasing trend in publishing articles related to dementia over time. In addition, dementia-related (modifiable) risk factors as principal news content was significantly associated with the year of publication. Despite a weak association between story categories and newspapers, the majority of articles reported preventive measures as the main story category. Although both newspapers featured more articles with a less negative tone across time when reporting on dementia, derogative wording, as discourse, was commonly used to address the illness. We have provided some insight into understanding how online newspapers potentially affect subjective representations of dementia as well as perpetuate dementia discourse. Finally, we suggest that future study may benefit from establishing a linkage between the depiction of dementia in online newspapers and the contextualization of dementia within cultures.

## 1. Introduction

The growing number of people diagnosed with various forms of dementia as a topical news item in the media often widely effects negatively among the general population [1]. Furthermore, a range of topics related to dementia has been featured and discussed explicitly in both print and in online newspapers, namely, what it is like to be diagnosed with dementia, the causes of dementia, available treatments and preventions, wording to address dementia, and available care services and how they operate. The online news title ‘Alzheimer vaccine passes crucial test’ [2] somewhat establishes a biomedical discourse and accelerates expectations among the general population. However, prioritizing biomedical aspects often obscures the relevance of the social impact of the news [3]. Evidence further showed that media coverage portrays individuals who develop dementia as ‘socially death’, as well as generates fear, invokes intense social anxieties, and social disconnectedness for the representing families [4,5].

Although a preventive discourse is common in the online news reporting of dementia, it often significantly ignores the experiences of people living with the disease [6]. A number of studies have argued that dementia-preventive behavior in news articles is more likely highlighted as brain health [7,8]. The presence of this discourse in news coverage is contradictory, which elevates implicit panic about the disease and often carries with it unintended destructive potential [9,10]. One could further argue that the representation of dementia in online newspapers might have a great influence on constructing our ideas and shaping our beliefs, ideologies, and perceptions surrounding the phenomenon [11]. 

Although the news media representation of mental illness has been studied since the 1970s, the representation of dementia and portrayal of people living with dementia (PlwD) is a relatively recent phenomenon [12]. Most of the recent studies related to the media’s representation of dementia and cultural narratives concluded that dementia is predominantly represented as a loss of self, crisis, war, and a living death [13,14]. Therefore, the representation of dementia in online newspapers and the discourse of dementia has become an important topic in order to address the societal perspectives of the disease. 

The aim of the present article is to investigate the possible effects of news consumption concerning dementia in online newspapers by comparing the content and form of news coverage on dementia by considering a five-year period of reporting. In addition, we explored how the dominant discourse of dementia might have been influenced by the circulated online news. This article addresses several aspects of news coverage related to dementia with a number of concerns that will further develop a critical but constructive insight into the cultural and social representation of dementia that exist in everyday life.

## 2. Online vs. Print Newspapers

The advent and advancement in technology has brought exciting new changes, as well as risk, for print newspapers in the 21st century [15]. The traditional way of seeking, gathering, processing, and producing news in the printed version has been replaced by online newspapers. This evolution indicates the emergence of new content creation and has provided a window into the construction of information on the internet. In other words, the new communication and interaction process with digitized news not only provides information but also invokes what to and how to think about it. 

As online newspapers have evolved, the use of news portals has increased significantly. Compared to printed newspapers, the global readership of online newspapers has increased 13% during 2015–2019, and of that, almost 43% of the readers share news content with other people [16]. A country-specific study found that the number of online news readership among the USA population has increased from 5% to 35% between 1995 and 2002 [17]. Despite an increased readership, online newspapers were criticized for simply reproducing the print product online. A number of studies further found an ambiguous relationship regarding the preference for a specific modality of newspapers.

Some recent studies, for example, reported that users devote more time and spend more effort while reading text online as compared to the readers of printed text [15,18]. Moreover, the increasing use of the internet and access to mobile devices and applications (known as ‘mobile apps’) have attracted print newspapers to adopt digitalized editions [18]. The online publications have advantages for user involvement, such as low barriers to entry, interactivity, and the timeliness of online news [19]. Readers today, thus, are no longer ‘passive’. They are a heterogeneous group with different perceptions who do not simply accept but interpret, reject, and challenge the news. The ‘easy to receive’ digitized information also influences the mode of communication and our understanding of the topics in the news [20]. However, the credibility and reliability of the information that can be approached easily from online newspapers has become a major problem, which has massive impacts on individuals and society. 

One study, on the other hand, identified that the same content of printed news still has a higher preference compared to the online format among the majority of readers [21]. One of the reasons is connected to the fact that the online newspapers might not have perceived some utility functions of the printed newspapers. Moreover, the mandatory requirement of an internet connection, incomprehensive information, the risk of news accuracy due to the demands of being the first to publish breaking news and the race against time are some of the major challenges of an online newspaper that can often be called into question [22].

Since online newspapers are one of the most quickly news-spreading platforms, they can widely influence and determine consumer behavior [21]. Thus, it is expected that online newspapers can be a strong means to facilitate negative perceptions that are associated with dementia, as well as to achieve the goal of raising public awareness of dementia [23].

## 3. Dementia and Discourse of Dementia

Dementia occurs across cultures but has been treated in a very specific way around the world. The disease has been widely connected to degeneration as a part of old age and positioned as one of the global public health priorities [24,25]. The pathological traits of dementia are thought to be universal that may vary in its manifestation and interpretation in other cultures [26]. Moreover, the experience and conceptualization of dementia might not carry the exact same meaning in Western and non-Western settings [27].

Simply, discourse is a system of thought or knowledge, and a formal way of communication that can be expressed through language (or words), and often constructs our experience of the surroundings. Thus, the discourse of dementia can be characterized as predominant words and images in health care literature, various media, public discourse, and policy [28,29].

Even though discussing the dominant discourse associated with dementia is the main goal of this study, we thought it would be beneficial for the reader to walk through current literature to articulate themselves about the general aim of this study. The following section provides a brief but informative overview of existing literature that highlighted dementia-related discourse in various contexts.

## 4. Brief Review of Relevant Literature

As Lupton [30] has argued:


*For many lay people, the mass media constitute one of the most important sources of information about health and medicine. Mass media portrayals contribute to the creation or reproduction of knowledges about illness and disease … they work to portray ill people in certain lights (for example, as ‘innocent victims’ or ‘deserving of their fate’).*


Emerging evidence supports that news coverage (both electronic and printed versions) creates and reflects public awareness of particular health conditions. It often constructs and mediates our understanding and plays a significant role in reproducing the cultural values and social perceptions of different diseases like dementia [6,7]. Lyons [31] argued that ‘dominant constructions of health and illness in the media may function to influence how people behave with regard to their own, and others’, health’. Adequate knowledge of dementia might help to decrease social anxieties among family members, caregivers, PlwD, as well as the general population, and enhance the awareness of preventive measures [32]. Therefore, in the process of knowledge production and building awareness of dementia across societies and individuals, the online news media has potential due to its easy accessibility that attracts a global audience with a widening demographic.

Scholars recently have been interested in studying the implications of news reports and how these reports reproduce the socio-cultural perceptions of dementia. The newspaper headlines on dementia are often characterized through phrases such as ‘violent invader’, ‘killer’, ‘merciless assault’, ‘dreaded’, and ‘cruel’ that inflict individuals’ physical pain and suffering [33]. In a recent study, Peel [13] found that dementia was represented as ‘worse than death’ in UK national newspaper articles. Peel further showed some other examples in news titles, such as, ‘The living death of Alzheimer’s’, and ‘Take a walk to keep dementia at bay’. These titles are paradoxical discursive representations of dementia (e.g., ‘panic/fear discourse’, ‘biomedical discourse’, ‘dementia worry’) that may shrink our thinking to consider other palliative care opportunities [34].

The conceptualization and the cultural meanings attached to dementia are diverse and are interpreted, embodied, or resisted differently by people in their respective social contexts [35]. Earlier studies investigated how dementia is marked as a socially constructed phenomenon [36], since people commonly believe that dementia is a normal part of aging [37,38]. One of the beliefs might have been connected to how dementia is represented in policy documents and in dominant discourse. Some of the overabundance of compelling policy document examples are “dementia poses one of the greatest societal challenges for the 21st century” [39], “21st century *plague*”, or “*war* on Alzheimer’s” [40], the *epidemic* taking over, the *tsunami* heading our way [29]. In a similar fashion, the way of depicting dementia in the dominant discourse is used to express the ‘othering’ discourse. Many family caregivers, for instance, believe there is nothing left of their beloved one [29], that the person with dementia is ‘socially death’ [5] and so on. 

However, raising public awareness of dementia has become an integral component in dementia strategies as well as in public health policies [41]. Countering negative rhetoric presentations, some inclusive understandings of dementia and positive attitude have emerged towards those living with the condition. Given that, the Alzheimer’s Disease International (ADI) and the World Health Organization (WHO) have proposed to raise awareness and knowledge of dementia to improve the care services for PlwD [39,42]. A similar recommendation has also been proposed as a main mechanism for improving the prevention and treatment of dementia [43]. In addition, various dementia groups advocate for a more open and neutral guideline to address PlwD. Some of the initiatives are Alzheimer Society of Canada’s ‘Person Centred Language Guidelines’ (2012), Alzheimer’s Australia’s ‘Dementia Language Guidelines’ (2014), the Alzheimer Society of Ireland’s ‘Dementia Friendly Language’ (2015), and the UK’s ‘Language Guide’ (2015) [28].

Although existing studies have provided insight into how stereotypes form social injustice for PlwD, researchers were grappling to strongly critique against the predominant medical discourse. A number of studies demonstrated a bias toward the clinical research of potential drug treatments [2,3,44]. Wilson and colleagues [45], for instance, showed that a prominent discourse of biomedical news reporting often highlighted dementia treatment as ‘breakthroughs’ or ‘cures’, which might be misleading or could lead to false hope among readers. Another recurring feature is biomedical colonization. It perpetuates dehumanized healthcare for PlwD through the practice of infantilization, intimidation, stigmatization, and objectification [29]. In particular, the news media exists to provide the pathology of the condition when featuring dementia in online news content. Consequently, the role of social and cultural factors is significantly absent and underinvested in stories about dementia in news coverage [6,33]. Thus, conducting research on the unanswered questions about how to determine how the social perceptions of dementia evolve through the influence of online news coverage might provide new insight.

## 5. Theoretical Considerations

Our understanding about dementia and those who live with the condition are often constructed and medicated by social media. The representation of dementia in media is a powerful tool that influences societal beliefs and individual behaviors. In this study, as media analysis, we took the opportunity to articulate three interconnected theoretical approaches: *cultivation theory, agenda-setting theory,* and *spiral of silence theory*.

*Cultivation theory* is a sociocultural theory which suggests that the media’s (e.g., television) representation of an issue can systematically shape individuals’ perceptions, beliefs, attitudes, and values [46]. The message conveyed by such media platforms affects viewer’s perceptions and creates a distortion of reality. More specifically, the storytelling function of the media, in turn, can influence both individual and societal attitudes (e.g., fear, desire for wealth) and behaviors (e.g., aggression, purchase behavior). Despite the early criticisms of the cultivation theory, the convergent findings of this theoretical approach have arguably strong testaments to examine cultural, social, and psychological impacts of the media.

Agenda-setting roles of the news media refers to the perceived importance of content in the news and the reproduction procedure of the public attention to specific topics at any particular point in time. The *theory of agenda-setting* comprises three core concepts: a media agenda, a public agenda, and the salience of news items and explicates how the news media (e.g., newspapers, television, news magazines) attribute attention to the specific issues to reproduce the underlying psychology of public opinion [47]. The consumers of news content not only learn about an issue but also use the information based on a news story. The more the news media emphasize a particular issue, the more people will rely on what they know about it to make their judgments [48]. In a similar vein, recent online news media demonstrate agenda-setting effects on public attitudes, opinions, and behaviors [49].

The *spiral of silence theory* describes the dynamics of public opinion in situations where people are afraid of being isolated or rejected by those around them and to which there is a pressure to conform in a context or society [50]. The spiral of silence process indicates peoples’ willingness or reluctance to express opinions in public settings if they felt that their opinions might not be widely shared. The mainstream news media (e.g., newspapers) can significantly influence this process. For example, congruence in media content, repetition of similar messages, and their presence as public sources of information can make differences in how people will process the information [51].

Despite criticisms, several studies have explicitly concluded that the relationships among the cultivation theory, the agenda-setting theory, and the spiral of silence theory are repetitive and cyclic [52,53]. Dementia in the news media, for example, is cultivated and overexposed to a certain manner and considered as a top agenda due to the importance given to it. It thus, is more likely to pursue the readers’ perceptions of dementia in a specific direction across time. In other words, the news media set agendas related to dementia for the public through continuous cultivation. Similarly, greater media coverage and favoring a one-sided view of dementia have an influence on interpersonal discussions (agenda-setting) that fit within the spiral of silence [53].

## 6. Current Study

Online news platforms potentially have a great influence on the issue of what differentiates online newspaper articles with printed articles regarding content and news stories. These newspapers have the characteristics of a print newspaper with the largest circulation but contain a wide variety of sections, such as news sections, discussion groups, and opinion sections that are made available to the readers. However, in this study, our aim was not to determine the extrinsic contextual factors, such as reading time, certain types of news and reasons for reading a newspaper [54]. Instead, we have conducted a comparative overview of the characteristics of coverage related to dementia in these online newspapers that might intensify our knowledge regarding the socio-cultural perceptions of the phenomenon. 

In this article, we argued that the portrayals and discourse of dementia are more of a concern of social and medical research catching up to be more sensitized to the necessity for change. Most of the pejorative and degrading perceptions of dementia could be abolished through consciousness raising, education, and lobbying by various advocacy groups. We further argued that using the dementia term from a social framework allows for a better understanding of the multiplicity of the disease in which the practices are situated (e.g., different socio-cultural contexts). Thus, a convenient and compassionate substitution of the word dementia is timely.

## 7. Study Aim and Research Questions

In this study, we investigated the temporal trends of news presentation and discourse concerning dementia in two online newspapers (*The New York Times*; NYT, and *The Guardian*; GDN) by comparing the *news content*: the subject matter or symbolic significance of news, the *news story*: a written content that informs the public about an event, concern, or idea, and the *overall tone*: news represents impartial, biased, and/or subjective meanings of the story of news coverage on dementia to consider a five-year period of reporting.

The depiction and prevalent contextualization of dementia in the news media is typically negative, which in turn impacts on public stigma, negative feelings, and social distance [55,56]. It further generates stereotypical images of PlwD that they are not fully human, are incompetent, and are burdens on society. These findings call for action to grow a public consensus on avoiding derogative language and to be mindful about the social impact of the words while addressing dementia and PlwD. Based on previous study findings on media analysis, we formulate a hypothesis in our study that there might be a linkage between the depiction of dementia in online newspapers and the contextualization of dementia within cultures.

To understand the hypothesis, three main research questions were posed:Are there changes in *news content* related to dementia in the NYT and the GDN over time?What are the differences in *news stories* between the NYT and the GDN across time?What is the *overall tone* in presenting topics related to dementia both in the NYT and the GDN?

The findings of this study would contribute to a better understanding of the role of online newspapers in influencing the development of social perceptions as well as public opinions on dementia and PlwD. 

## 8. Materials and Methods

A descriptive and content analysis undertaken in this study provided insight into the dementia-related news coverage by the two most-circulated online newspapers. This method typically comprises retrieving articles from a database, manually excluding articles less relevant to the research aim, constructing a structured coding frame, systematically coding each article, and performing statistical analysis of the coded data [57]. Several previous studies that compared newspaper reports between two countries justify the application value of this method [58,59,60].

### 8.1. Data Sources

Following the newspaper web rankings in 2019 (https://www.4imn.com/top200/ (accessed on 3 October 2019)), we selected the top two online newspapers. Online platforms were chosen as they are comparatively less expensive than printed periodicals, people can read them regardless of time and region, and the news can last for a longer period on the internet. The newspapers were selected based on their free accessibility and availability, broad-spectrum of readership, and high circulation records (see details www.websiteoutlook.com (accessed on 3 October 2019)) but separated from major printed international newspapers. 

Two online newspaper sources were:(1)*The New York Times* (https://www.nytimes.com/) (accessed on 11 December 2019). The online version of the newspaper was established in 1996. As of December 2019, the approximate daily average page viewers (readership) for this newspaper was 11.46 million per day.(2)*The Guardian* (https://www.theguardian.com/) (accessed on 11 December 2019). This newspaper launched the online version in 1995. As of December 2019, the approximate daily average page viewers (readership) for this newspaper were 8.74 million per day.

### 8.2. Selection of News Coverage 

A bracket of a 5-year timespan (November 2014 to November 2019) was implemented to generate a recent and broad dataset regarding the contemporary news on dementia in the NYT and the GDN, so it allows us to explore this phenomenon adequately. A 5-year timeframe was chosen to cover a growing public attention to aging in general and dementia specifically [61]. The study used the terms “Dementia” or “Alzheimer’s” for screening as they are often used synonymously in the news media [14]. Full texts of the articles were extracted, and comparisons were made between the newspapers.

### 8.3. Article Search and Manual Exclusion 

An initial screening was undertaken in order to test the methodology. This was essential to strengthening the research aim and to validate the sampling strategy for the main analysis [62]. The search term “Dementia” OR “Alzheimer’s” was used to identify the most specific articles containing dementia and PlwD-related news. The initial search retrieved 2543 articles (*n* = 1597 from the NYT; *n* = 946 from the GDN). A total of 388 articles (*n* = 107 from the NYT; *n* = 281 from the GDN) were screened manually based on inclusion and exclusion criteria. To be included, articles were required to have a title with dementia, to have been published during the study period, to be focused on issues related to dementia, and to have a news story which covers the social and clinical aspects of dementia. Articles were excluded if they did not mention dementia in the title and did not address PlwD. All articles were individually examined and 97 of them were excluded (25%), as they did not specifically discuss dementia. Following this step-by-step procedure, the 291 articles fulfilled all the selection criteria and were subsequently analyzed. In order to gather a broad but manageable dataset, the study considered all sections of the online newspapers to screen for articles that fit our scope.

### 8.4. Constructing the Coding Design

We clustered the timeline of collected data in 5 consecutive years where ‘Year 1’ refers to ‘November 2014 to October 2015’ and ‘Year 5’ refers to ‘November 2018 to November 2019’. A comprehensive coding framework was developed (see Appendix A) based on previous literature published in assessing media discourse, mental illness, as well as other diseases in newspaper coverage [6,63,64,65]. 

The following factors and characteristics in the newspapers were considered in the coding procedure:

#### 8.4.1. Title Focus 

Departing from the concept of ‘personification’ described by Lebowitz & Ahn [66], this coding scheme was developed by considering the focus area of the title in the newspaper. Personification was defined as whether the title discloses personal details or described dementia as a powerful biomedical challenge (e.g., “Dementia is too big a problem to walk away from—for Pfizer or any of us”), fearsome or dreadful, (e.g., “Dementia is the plague of our time, the disease of the century”), and as a stigmatized element (e.g., “Robin Williams’s widow points to dementia as a suicide cause”) [3,65]. Examples of coded titles that focused on the individual (diagnosed/caregiver) were “Can you tell that Paul, 58, has dementia? No one expects it in a fit, middle-aged man” and “Dementia damaged Ron’s emotions. Then I took him chocolate shopping”.

#### 8.4.2. News Section

All articles in the two online newspapers appeared in 17 different sections where 3 sections (business, opinion, sports) were overlapping. The full text of the articles was examined individually and clustered based on the best matched 4 sections coded as culture, lifestyle, news, and opinion. For example, the title ‘Cultural awareness improves dementia care for South Asian minorities’ (in the GDN) published in the ‘Other’ section of the newspaper was included in ‘Culture’; the title ‘The Right Kind of Exercise May Boost Memory and Lower Dementia Risk’ (in the NYT) published in the ‘Mind’ section of the newspaper was included in ‘Lifestyle’; the title ‘Pollution leads to greater risk of dementia among older women, study says’ (in the NYT) published in the ‘Other’ section of the newspaper was included in ‘News’; the title ‘Should I Get Involved in Helping a Neighbor With Dementia?’ (in the NYT) published in the ‘Magazine’ section of the newspaper was included in ‘Opinion’. A consensus was made among the authors of this study to determine these four codes while merging news titles published in other sections of the newspapers. This process helped us to keep an allied and common distribution of the news articles.

#### 8.4.3. Principal Content

The contents of full text articles published in different sections of the newspapers were coded into 5 major groups with sub-groups as ‘Care’ (formal care, informal care, social care, and self-care, e.g., “Integrated care is key for dementia patients”), ‘Information’ (attitude, awareness, cost, policy, e.g., “Researchers turn to crowdfunding to develop Alzheimer’s drug”), ‘Risk factor’ (modifiable, non-modifiable, e.g., “The right kind of exercise may boost memory and lower dementia risk”), ‘Sign’ (early, late; e.g., “Long-winded speech could be early sign of Alzheimer’s disease, says study”), and ‘Treatment’ (biomedical, technological; e.g., “Biogen reports its Alzheimer’s drug sharply slowed cognitive decline” or “Blood-thinning drugs can reduce risk of dementia by up to 48%”). These codes were developed following the WHO’s *Global action plan on the public health response to dementia 2017–2025* that comprises 7 action areas (1. Dementia as a public health priority 2. Dementia awareness and friendliness, 3. Dementia risk reduction, 4. Dementia diagnosis, treatment, care, and support, 5. Support for dementia caregivers, 6. Information systems for dementia, 7. Dementia research and innovation) [67].

#### 8.4.4. News Story Category

As a fundamental component of the study, news story categories in the newspapers were clustered into 4 different codes as burden, knowledge, prevalence, and prevention through determining whether the story dealt with any of these issues.

An example of burden is “Family’s lonely fight over dementia hospital death”. Examples of knowledge are “Seven ways to help avoid dementia” and “What is Alzheimer’s disease?”. An example of prevalence is “1.2 million people in England and Wales will have dementia by 2040”. An example of prevention is “Dementia: Eat better, exercise, and reduce smoking and drinking to cut risk”.

#### 8.4.5. Overall Tone

Both the titles and full text articles were classified into 4 codes according to their tones—positive, negative, neutral, and mixed—based on the overall news presentation demeanor. A positive code refers to the title and articles depicting progressive, optimistic, and hopeful presentations, such as “The right kind of exercise may boost memory and lower dementia risk”. An example of a negative code is “Dementia’s gift: Facing cancer without the fear”and represents fearful characteristics of dementia. A neutral code is attached to titles or articles that present impartial news such as “Shops, cafes and round-the-clock care: life in a dementia village”. News titles and articles containing both favorable and negative elements were labeled as mixed, such as “Forgetting but not gone: Dementia and the arts”.

### 8.5. Inter-Rater Reliability

During the screening stage, 30% of the total articles were independently screened by two researchers and the inter-rater reliability of coding was assessed using percentage agreement. The inter-rater reliability with kappa score ranged from 0.68 to 0.95. According to Landis and Koch [68], kappas in the 0.41–0.60 (moderate agreement); 0.61–0.80 (substantial agreement); and above 0.80 (almost perfect agreement).

### 8.6. Statistical Analyses

The statistical analysis of the data was conducted by the Statistical Package for the Social Sciences (SPSS) software (v 20.0; IBM Corporation, Armonk, NY, USA), with a *p*-value of < 0.05 indicating statistically significant. Categorical variables were coded while frequencies and percentage distributions were used as descriptive analysis methods.

### 8.7. Ethical Considerations

All the data sources were publicly available. Hence, the ethics committee approval was not required for this study.

## 9. Results

The majority of articles related to dementia were published in the GDN (*n* = 209, 72%) during the study period (Table 1). In the NYT, the proportion of published articles slightly increased from Year 1 (20%) to Year 5 (27%), whereas a considerably opposite trend was observed for the GDN (Year 1, 33%; Year 5, 16%). Table 1 further shows that on average nearly 3 quarters (the NYT, *n* = 74; the GDN, *n* = 139) of the articles in both newspapers underlined dementia (the illness) as compared to titles described about individuals with dementia diagnosis or their caregivers (the people) over the study period. Regarding the news sections, the highest number of articles published for the NYT was in the *Lifestyle* section (66%), while it was in the *News* section (66%) for the GDN. The *Culture* section was deemed as of comparatively less importance in both newspapers (the NYT, 5%; the GDN, 8%).

### 9.1. Principal Content

Comparing the two newspapers, the NYT prioritized risk factors over other news contents whereas dementia-related information led the GDN (Figure 1A). Other than that, the proportion of news content regarding all categories in both newspapers was similar. Results further showed a statistically significant association between the news content and the year of publication (*p* < 0.002; *X^2^* = 37.14; *df* = 16). 

By analyzing the principal content of both newspapers across time, the proportion of content about modifiable risk factors almost doubled in Year 5 (29%) compared to Year 1 (15%). On the other hand, the content on information decreased dramatically over the study period (Year 1, 43%; Year 5, 13%). While dementia-related signs were poorly reported in Year 5, the content on care showed mixed results (decreased until Year 3 then increased, Figure 1B).

### 9.2. Story Category

The proportion of story categories showed a different result in both newspapers. For example, the NYT was more likely to report on preventive measures (33%) compared to other story categories. On the other hand, a similar high proportion of coverage was observed on dementia-related knowledge (80%) and prevalence (83%) in the GDN, as compared to burden and prevention. Surprisingly, a considerable proportion of stories appeared to indicate dementia as a burden in both newspapers (the NYT, 28%; the GDN, 72%) (Figure 2A). However, no major association was observed between story categories and newspapers.

In Year 1, dementia-related burden and knowledge received more attention, while a decreasing trend was observed in the subsequent years (Years 2, 3, 4, and 5) (Figure 2B). Compared with Year 1 and Year 2 (28%), news stories were less likely to describe the prevalence of dementia in Year 4 (17%) and Year 5 (6%). However, the proportion of news stories related to prevention increased over time from Year 1 (17%) to Year 5 (23%). 

### 9.3. Overall Tone

Our results showed that the NYT published more articles with a negative tone (35%) and less with a mixed tone (14%) (Figure 3A). On the other hand, in the GDN, articles with a mixed tone received more importance (86%) while a considerable proportion of articles were featured with a negative tone (65%). Both newspapers showed a similar trend in publishing articles with positive and neutral tones. No statistically significant association was observed between the articles’ overall tone and the newspapers.

Figure 3B displays the overall tone of published articles combining title and full text across time. Although the proportion of news reflecting a negative tone decreased from Year 1 (31%) to Year 5 (19%), news with a positive tone was almost stable during the study period. Notably, news articles were more likely to have either positive or neutral tones over the study period while representing dementia. Despite a slight increase of neutral tones (from Year 1, 18% to Year 5, 21%), the proportion of articles with mixed tones decreased dramatically in Year 5 (11%) compared to Year 1 (46%).

### 9.4. Representation of Dementia in Newspapers

Dementia has mostly been depicted with derogative wording in both newspapers. The result showed that dementia as an illness or people with the condition were often equivalent to negative imagery language. The representation of dementia in news titles was ‘It will take more than scientific brilliance to win the battle against dementia’, ‘Dementia is the plague of our time, the disease of the century’ (the GDN); ‘Doctors need to talk to families about guns and Dementia’ (the NYT). On the other hand, people living with dementia have also been portrayed in a similar vein. Such as ‘Research shines light on why women more likely to develop Alzheimer’s’, ‘Dying well when you have dementia’ (the GDN); ‘Dementia patients fuel assisted living’s growth. Safety may be lagging’, ‘Dementia may never improve, but many patients still can learn’ (the NYT).

Besides these, some of the frequent words in the news titles directed to dementia were ‘Pain’, ‘Threat’, ‘Financial ruin’, ‘Abuse’, ‘No longer safe’, ‘Tragedy’, ‘Lonely fight’, ‘Devastating’, ‘Fight for dignity’, ‘Isolation’, ‘Time bomb’, ‘Heist’.

However, some positive language (wording) was also observed in the news titles, such as ‘Music in dementia care sounds promising, but there is a catch’ ‘Make a playlist for someone with dementia: the results will astonish you’ ‘Art can be a powerful medicine against dementia’ (the GDN); ‘Alzheimer’s patients keep the spark alive by sharing stories’, ‘Studies confirm brain plaques can help predict AD’ (the NYT).

## 10. Discussion

We know quite a lot about the representation of dementia in the news media. However, the coverage of dementia, especially in online newspapers and exploring shifts over time, is relatively less known. The portrayal of the disease by describing news content, news stories, and overall tone in articles published in two online newspapers is the unique contribution of the current study to the literature.

Overall, we found a decreasing trend in article publications related to dementia from Year 1 to Year 5. This pattern illustrates that dementia is indeed receiving less attention in recent times than before, which partially corresponds to the other study [6]. However, a recent study reported an increase in the number of articles containing dementia until 2015 [65]. Getting less attention might have a two-fold consequential influence on formulating public opinion associated with dementia: first, a common misapprehension of dementia as a normal part of the aging process could be persistent with entrenched stigma; second, personal antipathy toward dementia could increase and hamper the efforts to widen social inclusion of the population with this condition.

The majority of the news coverage included in this study focused on the illness rather than on the individuals diagnosed with dementia or their caregivers. Since both newspapers have a high readership, one of the consequences of highlighting the disease might sculpt to present a particular attitude or belief on dementia by obscuring some other actors in society. It could further lead our understanding to a narrower route of care and reproduce dominant as well as tragedy discourse. For instance, a negative stereotypical attitude might be elevated at both individual and societal levels toward the disorder. The title from the GDN can be a better example, ‘Dementia is the plague of our time, the disease of the century’. It is also possible that some news reports overestimate the caregivers’ burden. Some corresponding titles are: ‘I have to treat patients like objects: the harsh reality of working in dementia care’ (the GDN); ‘Living with the patients I’m losing to AD’ (the NYT). It can thus be further suggested that both newspapers are more likely to report on the care need or caregiving burden of dementia, which in turn impacts on the dementia-related social stigma [56]. Consequently, help-seeking behavior may be delayed, healthcare professionals may be reluctant to give a dementia diagnosis, and the overall human rights of people with dementia may be violated [69,70].

There is a growing recognition that a healthy lifestyle is associated with a substantially lower risk of dementia that may protect, prolong, or even prevent against the diagnosis and might lead to a longer healthy life [71]. Results in our study showed that, in the NYT, a major proportion of the articles dealing with dementia appeared in the *Lifestyle* section. This result indicates the importance of our day-to-day life activities. Some corresponding titles are: ‘Dementia may never improve, but many patients still can learn’; ‘A 1-hour walk, 3 times a week, has benefits for Dementia’ which relates to biomedical discourse on ‘living well’ with dementia. In addition, there were a number of articles advocating for various, daily life practices that might prevent dementia. Some of the titles are: ‘Healthy lifestyle may reduce Dementia risk’; ‘Smoking may increase Dementia risk’ that establish ‘controllable’ aspects of dementia. Lyons [31] argued that this representation is an extension of liberal-humanistic discourse where the individual is responsible for his or her own health and wellbeing. In contrast, in the GDN, the proportion of articles increased slightly mainly in the *Lifestyle*, *Culture*, and *Opinion* sections over time but decreased in the *News* section over time. 

With minimal reporting on the subjective experiences of dementia, both newspapers are more reluctant to enter the discussion on ‘what it is like to live with dementia’. For example, ‘Mum has dementia and now Dad’s dead she will have to sell her home. Why?’ (the GDN); ‘Fraying at the edges: Her fight to live with AD’ (the NYT). Both stories are told either by the daughter or by the caregiver, where the voices of the individuals with lived experience were ignored. Other people are making choices on their behalf. In such ways, it is possible to strengthen our belief in ‘what people with dementia cannot do’ rather than ‘what they can do’. Highlighting the latter discourse, Cowdell [72] suggested that research can be conducted ‘with’ people with dementia rather than ‘on’ them. We argue that if future dementia research wants to make a positive impact on the lives of people with dementia, the individual’s lived experience should be in focus, not just the disease. It is further important to know in what way individuals’ social identity (e.g., based on age, gender, class, race, and ethnicity) shapes their experiences and perspectives of dementia.

Our results showed that the modifiable risk factors related to dementia received a great deal of attention in recent times. One possible reason is related to the lack of a cure for dementia and the modest impact of pharmacological treatment of dementia [73]. A recent study detected nine major modifiable risk factors that can reduce 35% of dementia incidence [74]. Therefore, it can be suggested that making changes in specific aspects of lifestyle may influence the delaying of a dementia diagnosis. Earlier studies supported the argument that a healthy lifestyle or risk-reduction strategies have potential benefits to reduce dementia risk and have less adverse effects [75,76]. 

Notably, there was limited content on the cost of dementia care and the current allocation of funding for research mostly informed about the budget cut in this field. For example, ‘NHS to discontinue dementia diagnosis payments to GP practices’ (the GDN); ‘Costs for Dementia care far exceeding other diseases, Study finds’ (the NYT). Moreover, the under-reported content on the early or late signs of dementia was common in both newspapers with exceptions such as ‘Long-winded speech could be early sign of Alzheimer’s disease, says study’ (the GDN); ‘Prolonged sleep may be early warning sign of dementia’ (the NYT). The lack of a positive portrayal of dementia in the news coverage influences and reflects cultural stigma, which can hinder public awareness of dementia [77]. 

Despite the increase in news stories related to preventive measures, overstressing on dementia burden will significantly elevate psychological disorders, such as stress, depression, and anxiety among caregivers [78]. On the other hand, highlighting the potential preventive measures are promising strategies to prolong the disease progression and might be an advantageous tool to mediate the social awareness of dementia [74]. While no statistically significant differences were found between story categories and year of publication, the readers’ perspectives on dementia might not be altered or changed by these story categories.

An earlier comparative study between two online newspapers found a neutral tone in the majority of the articles over time that differs from our findings [65]. We found a higher proportion of articles with a negative tone in both newspapers analyzed in our study. It indicates an overt practice of negative representation of dementia in news reporting, which can significantly perpetuate the stigma of dementia among the public, healthcare professionals, and people with dementia [56]. For example, ‘Dementia poses threat to health similar to HIV and Aids’ (the GDN); ‘Hospital no place for those with dementia’ (the GDN); ‘Dementia’s gift: Facing cancer without the fear’ (the NYT). We argue that these negative coverages require attention both from individuals as well as societal levels. On the other hand, a declining trend in negative tones was observed over time, suggesting an increased awareness and depicting a more positive framing of the disease. Some corresponding titles are: ‘When I was diagnosed with dementia, I thought it was the end of the world. It’s not’ (the GDN); ‘Alzheimer’s patients keep the spark alive by sharing stories’ (the NYT). These types of coverage may reinforce the healthcare providers to think about the abilities and agencies of people with dementia.

## 11. Potential Implications

This study contributes to formulating an effective understanding about dementia to larger audiences along with several possible implications. Firstly, the results show that the representation of dementia in news coverage is negatively stereotyped against people with the condition. News coverage of this nature could perpetuate the fear and stigma of dementia in society. The fear of dementia and discriminatory attitudes toward PlwD could affect help-seeking behavior. Nonetheless, content emphasized on modifiable risk factors of dementia will raise public awareness to maintain a healthy lifestyle.

Secondly, the overrepresentation of the dementia care burden in news stories could accelerate both social and cultural stigmas in every level of society. An alternative model of care, such as dementia-supportive communities and person-centered care would support independent living for PlwD.

Lastly, the discourse of dementia depicted in news coverage requires serious re-evaluation. The use of dementia-friendly wording while addressing PlwD could be an effective tool. A consistent guideline of addressing PlwD both in media and public policy can be developed incorporating health care professionals, family members of PlwD, and the individuals diagnosed with dementia.

## 12. Limitations

We acknowledge several limitations of our study. There is a risk of generalizing the perspectives of dementia in the two countries due to probable cultural differences. The time period and the number of newspapers we have examined in this study were relatively short. Thus, the results might not be generalizable for other periods. Although the newspapers that we have chosen have higher readership, the results can differ from other online newspapers. We also excluded visual images, videos, podcasts, and other animations in our analysis that might provide additional insight. Despite these limitations, the findings in our study contributed to the body of knowledge related to the representation and discourse of dementia in news coverage. Comparing a British newspaper with a newspaper published in the USA further provided insight into having a more international impact of dementia-related news coverage. 

## 13. Conclusions

Our findings identified that news articles related to dementia have declined in general during the study period, while the proportion of articles in different sections has increased. Moreover, the decreasing trend in publishing negative articles on dementia across time signals a positive societal change. This would be a gateway for policy makers if they want to reflect specific changes in online newspapers. The representation of dementia based on medical perspectives is important; however, neglecting other social perspectives and aspects of life with dementia could lead to the dehumanization and exclusion of PlwD. Future research may be able to link the depiction of dementia in online newspapers directly to measures of societal changes, discourse of dementia, and dementia stigma over time. One of the main issues should be how the media’s representation of dementia is translated to the general public and different social groups. The contextualization of dementia within cultures would be helpful to understanding the impacts of dementia portrayal in the news media and how the portrayal is reflected on attitudes, feelings, and intended or actual behaviors toward PlwD. We hope our study may help to redefine the collective perceptions of dementia and that detailed exploration of news coverage can advocate more attention on people with the condition.

## Figures and Tables

**Figure 1 ijerph-18-10539-f001:**
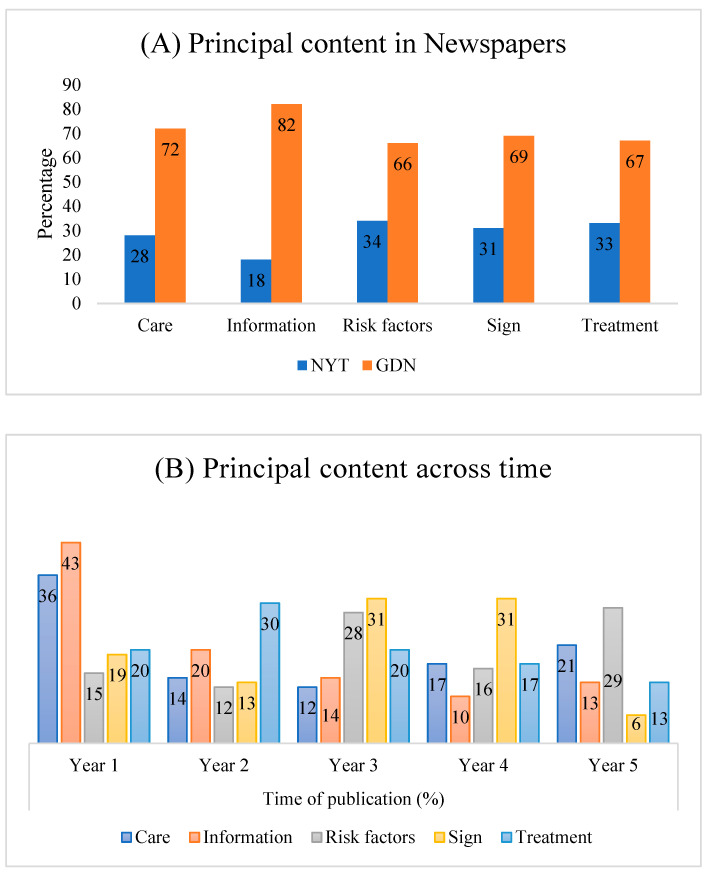
Principal content in NYT and GDN from Year 1 to Year 5. (**A**) Principal content in NYT and GDN, including care, information, risk factors, signs, and treatment. (**B**) Principal content of different categories across time from both NYT and GDN.

**Figure 2 ijerph-18-10539-f002:**
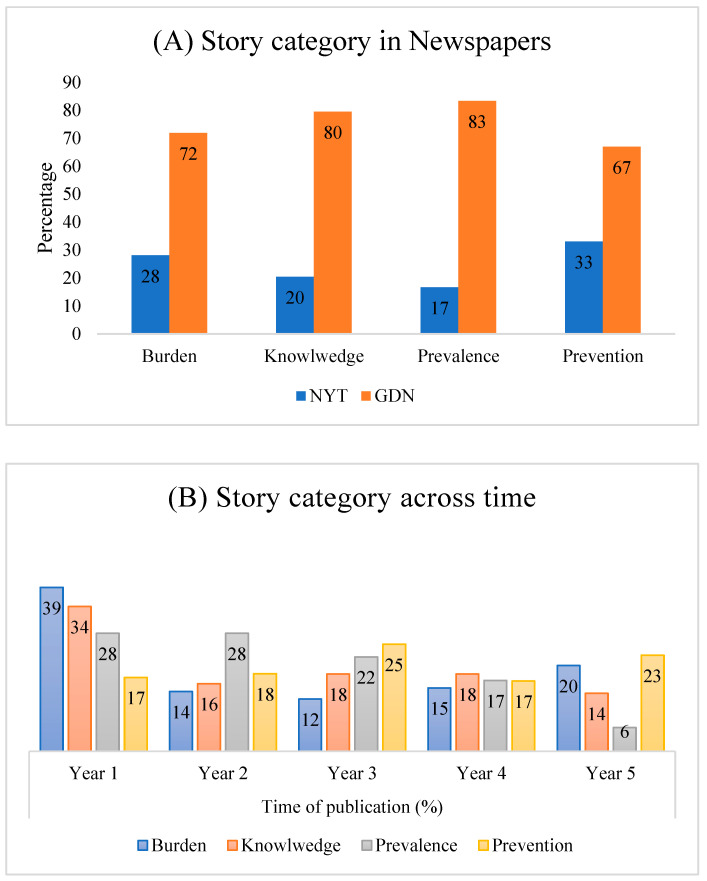
Story category in NYT and GDN from Year 1 to Year 5. (**A**) Story category in NYT and GDN, including burden, knowledge, prevalence, and prevention. (**B**) Story category of different categories across time from both NYT and GDN.

**Figure 3 ijerph-18-10539-f003:**
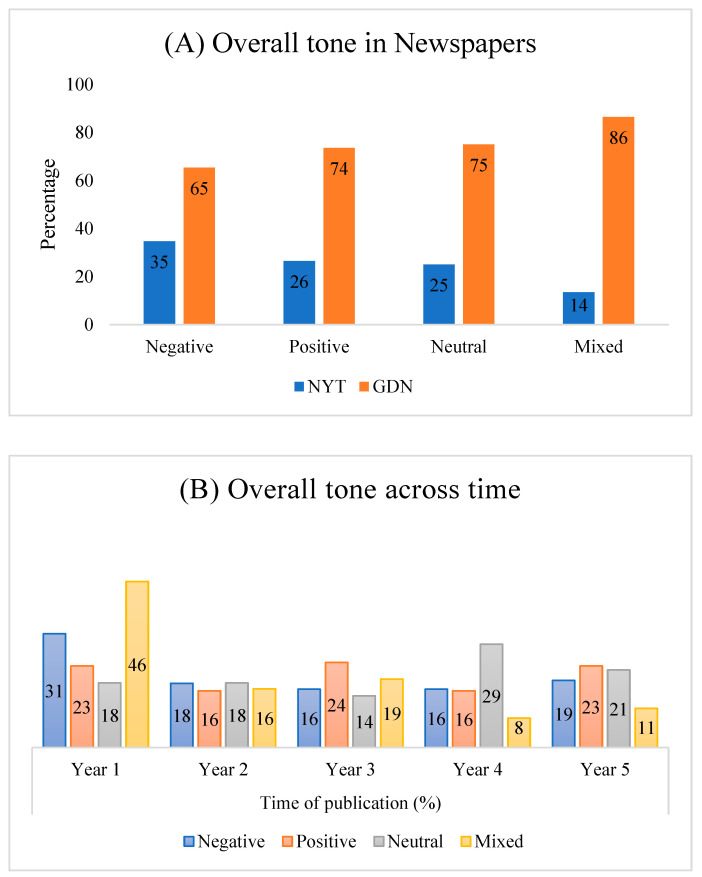
Overall tone in NYT and GDN from Year 1 to Year 5. (**A**) Overall tone in NYT and GDN, including negative, positive, neutral, and mixed. (**B**) Overall tone of different categories across time from both NYT and GDN.

**Table 1 ijerph-18-10539-t001:** Yearly distribution of published articles in relation to title focus and news section in NYT and GDN.

	NYT *n*(%)						GDN *n*(%)					
	Year 1	Year 2	Year 3	Year 4	Year 5	Total	Year 1	Year 2	Year 3	Year 4	Year 5	Total
Published articles	16(20)	19(23)	11(13)	14(17)	22(27)	82(28)	68(33)	30(14)	44(21)	33(16)	34(16)	209(72)
*Title Focus*												
Dementia	16(100)	17(89)	11(100)	11(79)	19(86)	74(90)	46(68)	19(63)	31(70)	21(64)	22(65)	139(67)
Person *	0(0)	2(11)	0(0)	3(21)	3(14)	8(10)	22(32)	11(37)	13(30)	12(36)	12(35)	70(33)
*News Section*												
Culture	1(6)	1(5)	0(0)	2(14)	0(0)	4(5)	8(12)	7(23)	0(0)	1(3)	1(3)	17(8)
Lifestyle	9(56)	14(74)	8(73)	7(50)	16(73)	54(66)	11(16)	1(3)	6(14)	6(18)	3(9)	27(13)
News	4(25)	3(16)	3(27)	3(21)	5(23)	18(22)	39(57)	20(67)	34(77)	19(58)	26(76)	138(66)
Opinion	2(13)	1(5)	0(0)	2(14)	1(5)	6(7)	10(15)	2(7)	4(9)	7(21)	4(12)	27(13)

* diagnosed/caregivers.

## Data Availability

Publicly available datasets were analyzed in this study that can be found in Newpaper’s website. The dataset can also be available on request from the corresponding author.

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
