# Peer review of "Dementia in Media Coverage: A Comparative Analysis of Two Online Newspapers across Time"

_ijerph, 2021, doi:10.3390/ijerph181910539_

Round 1

Reviewer 1 Report

The article concerns an exciting topic but contains methodological deficiencies and inadequacies. I don't feel compelled to argue with the values in the tables, but many questions arise.
1 The article is not sufficiently embedded in the literature on media studies and media influence. First of all, reference should be made here to such theories as George Gerbner's cultivation theory, McCombs and Shaw's agenda-setting theory, the spiral of silence theory proposed by the German political scientist Elisabeth Noelle-Neumann. Without this, the theoretical part is incomplete. Embedding in these theories would also help to deepen the conclusions;
2. the authors resign from formulating research hypotheses and do not argue it in any way. Yes, a hypothesis is not an idol. It is possible to write an empirical article of good quality without formulating a hypothesis - but this should be justified (e.g. by the specificity of the methods used).
3. the relationships and dependencies "print papers vs online papers" are much more complicated than the authors see. On the other hand, I understand that this is not the primary research thread of the article, and perhaps such a brief highlighting of the problem is sufficient. However, it is essential to remember that online websites of papers often are different in content from print editions. In the era of Internet development, newspapers have also gained a tool for promotion and interaction, which social media are. There are more such nuances, but I don't think requiring the authors to elaborate on this point makes sense. However, I encourage them to reflect on it - is it possible to develop it somehow?
4. the authors compare a British newspaper with a newspaper published in the USA. I understand that both are in English and reach an international audience. Nevertheless, the perspective of particular issues may be slightly different in the two countries. Aren't the authors afraid that this might affect the results of the study? Do they happen to be comparing inherently incomparable things?
5 The methodology is insufficiently described. We do not know whether the authors intended to conduct a Critical Discourse Analysis or Content Analysis.

Author Response

Dear Dr. Shero Han and Reviewer,

Thank you and the reviewers for the insightful and helpful comments on our manuscript entitled " Dementia in media coverage: A comparative analysis of two online newspapers across time" (Manuscript ID: ijerph- 1367434). We have substantially revised the manuscript to address all of the insightful and probing comments with the inclusion of extensive new text aimed at clarifying the reviewers’ points.

Below, the reviewers’ comments are in normal font, and our responses are in red. In the revised manuscript, significant changes are in track changes mode.

Response to Reviewer 1 Comments

Comments and Suggestions for Authors

The article concerns an exciting topic but contains methodological deficiencies and inadequacies. I don't feel compelled to argue with the values in the tables, but many questions arise.

1 The article is not sufficiently embedded in the literature on media studies and media influence. First of all, reference should be made here to such theories as George Gerbner's cultivation theory, McCombs and Shaw's agenda-setting theory, the spiral of silence theory proposed by the German political scientist Elisabeth Noelle-Neumann. Without this, the theoretical part is incomplete. Embedding in these theories would also help to deepen the conclusions;

Response 1: Thank you for pointing this out. We have added a theoretical consideration section in the manuscript as per your suggestions. This was really helpful to improve the quality our manuscript.

  1. the authors resign from formulating research hypotheses and do not argue it in any way. Yes, a hypothesis is not an idol. It is possible to write an empirical article of good quality without formulating a hypothesis - but this should be justified (e.g. by the specificity of the methods used).

Response 2: This is a very important point. We have included a hypothesis in the aim section.

  1. the relationships and dependencies "print papers vs online papers" are much more complicated than the authors see. On the other hand, I understand that this is not the primary research thread of the article, and perhaps such a brief highlighting of the problem is sufficient. However, it is essential to remember that online websites of papers often are different in content from print editions. In the era of Internet development, newspapers have also gained a tool for promotion and interaction, which social media are. There are more such nuances, but I don't think requiring the authors to elaborate on this point makes sense. However, I encourage them to reflect on it - is it possible to develop it somehow?

Response 3: This is an important suggestion. We have added the suggested description about online vs printed newspaper and wrote some reflections on it.

  1. the authors compare a British newspaper with a newspaper published in the USA. I understand that both are in English and reach an international audience. Nevertheless, the perspective of particular issues may be slightly different in the two countries. Aren't the authors afraid that this might affect the results of the study? Do they happen to be comparing inherently incomparable things?

Response 4: We improved the discussion about the newspaper in different countries and added some references on why it’s OK to study newspaper from different countries. We also improved the study limitation which justified our study limitations on perspectives of dementia in two countries.

5 The methodology is insufficiently described. We do not know whether the authors intended to conduct a Critical Discourse Analysis or Content Analysis.

Response 5: This is an important suggestion. The methodology section has been revised substantially highlighting analysis method and its importance on our study.

Reviewer 2 Report

A very interesting paper. I found the literature adequate and also the introduction provide sufficient background. 

The weakest point of this research is the number of newspapers that were studied (only two). However, the authors acknowledge this limitation and I believe the results could be used as a base for future research in this field.

I would suggest the following minor, changes.

- In the methodology section the author(s) should clarify if the data collection (screening) was made manually or automatically using software and a web scraper.

- I would suggest developing a bit more on the impact of this research in the conclusions.

Other than that, I believe the paper merits publication.

Author Response

Dear Dr. Shero Han and Reviewer,

Thank you and the reviewers for the insightful and helpful comments on our manuscript entitled " Dementia in media coverage: A comparative analysis of two online newspapers across time" (Manuscript ID: ijerph- 1367434). We have substantially revised the manuscript to address all of the insightful and probing comments with the inclusion of extensive new text aimed at clarifying the reviewers’ points.

Below, the reviewers’ comments are in normal font, and our responses are in red. In the revised manuscript, significant changes are in track changes mode.

Response to Reviewer 2 Comments

Comments and Suggestions for Authors

A very interesting paper. I found the literature adequate and also the introduction provide sufficient background.

The weakest point of this research is the number of newspapers that were studied (only two). However, the authors acknowledge this limitation and I believe the results could be used as a base for future research in this field.

I would suggest the following minor, changes.

- In the methodology section the author(s) should clarify if the data collection (screening) was made manually or automatically using software and a web scraper.

Response 1: This is an important suggestion. The methodology section has been revised substantially with detail screening process, data analysis method, and the importance of the method in our study.

- I would suggest developing a bit more on the impact of this research in the conclusions.

Other than that, I believe the paper merits publication.

Response 2: We added potential implications of our study as a new section as per your suggestion and also reflect on it in the conclusion.

Round 2

Reviewer 1 Report

The corrections made by the authors have improved the quality of the article enough to be published.